# Non-Conventional Sucrose-Based Substrates: Development of Non-Dairy Kefir Beverages with Probiotic Potential

Pedro Paulo Lordelo Guimarães Tavares [1], Clariane Teixeira Pessoa Mamona [1], Renata Quartieri Nascimento [2], Emanuele Araújo dos Anjos [3], Carolina Oliveira de Souza [1], Rogéria Comastri de Castro Almeida [4,*], Maria Eugênia de Oliveira Mamede [1] and Karina Teixeira Magalhães-Guedes [1]

1   Post-Graduate Program in Food Science, Federal University of Bahia (UFBA), Rua Barão de Jeremoabo 147, Campus Ondina, Salvador 40170-115, Brazil; pp.lordelo@gmail.com (P.P.L.G.T.); clarianepessoa@hotmail.com (C.T.P.M.); carolods@ufba.br (C.O.d.S.); mmamede@ufba.br (M.E.d.O.M.); karynamagat@gmail.com (K.T.M.-G.)
2   Post-Graduate Program in Biotechnology/RENORBIO—Institute of Health Sciences, Federal University of Bahia (UFBA), Avenida Reitor Miguel Calmon s/n, Campus Canela, Salvador 40231-300, Brazil; requarti@hotmail.com
3   Pharmacy School, Federal University of Bahia (UFBA), Rua Barão de Jeremoabo 147, Campus Ondina, Salvador 40170-115, Brazil; maanuaaraujo@hotmail.com
4   Food Science Department, School Nutrition Federal University of the Bahia (UFBA), Rua Basílio da Gama s/n, Campus Canela, Salvador 40110-907, Brazil
*   Correspondence: rogeriac@ufba.br

**Abstract:** There is a scarcity of studies evaluating the influence of different commonly marketed sugars in water kefir beverage production. Therefore, this study aimed to evaluate the fermentation of water kefir grains in different sugary solutions: brown, demerara, refined, coconut, and cane molasses. A total of 10% of each type of sugar was dissolved in sterile water to which 10% of kefir grains were then added and fermented for 48 h at room temperature. Analyses of pH/acidity, soluble solids, lactic/acetic acids, and lactic acid bacteria and yeast counts were performed, in addition to grain weighing at 0 h, 24 h, and 48 h. The microbial biodiversity was measured using PCR-DGGE and DNA sequencing at the species level. A sensory acceptance test was performed on all beverages. *Lactobacillus*, *Lacticaseibacillus*, *Lentilactobacillus Lactococcus*, *Leuconostoc*, *Acetobacter*, *Saccharomyces*, *Kluyveromyces*, *Lachancea*, and *Kazachstania* were present in the kefir grains and the beverages. Molasses showed a more intense fermentation, with greater production of organic acids and higher lactic/acetic acid bacteria and yeast counts (7.46 and 7.49 log CFU/mL, respectively). Refined sugar fermentation had a lower microbial yield of lactic/acetic acid bacteria (6.87 log CFU/mL). Smith's salience index indicates that the brown-sugar kefir beverage was better accepted among the tasters. The results indicate that the use of alternative sources of sugar to produce water kefir beverages is satisfactory. This opens up new perspectives for the application of kefir microorganisms in the development of beverages with probiotic and functional properties.

**Keywords:** molasses; brown sugar; non-dairy kefir; water kefir; lactic acid bacteria; yeast; lactic acid

## 1. Introduction

Currently, consumer awareness of natural/healthy food consumption has increased significantly. Therefore, there has been great interest in the development of new types of functional foods/beverages with probiotic potential [1,2]. Moreover, consumers are interested in non-dairy diets due to the allergic effect of dairy products on some consumers [3]. As an important source of probiotic microorganisms, water kefir is a good example of a non-dairy food/beverage [1–3].

Kefir is a beverage that originated in the Caucasus Mountains region, the border between Europe and Asia [1]. Kefir can be considered a probiotic beverage because it

contains microorganisms that help to maintain a healthy intestinal microbiota [2,3] and its consumption has several positive effects on human health, such as cholesterol and blood-glucose control and antihypertensive and anti-inflammatory potential, among others [3]. Water kefir is produced when kefir grains are grown in a solution containing sugar and water [4]. Thus, it is a dairy-free probiotic beverage [5]. Once the fermented product has been filtered, the grain-free beverage is known as "water kefir" or "sugary kefir" depending on the country [1,3]. "Tibico" or "Tibetan mushrooms" are other names frequently used [4]. Water kefir grains consist of a polysaccharide matrix, kefiran (mainly dextran), in which microorganisms are embedded [1–5]. The yellowish water kefir grains are jelly-like and translucent in appearance, with irregular shapes and sizes ranging from millimeters to a few centimeters [3–5] (Figure 1). Figure 1 shows water kefir beverage production: water kefir grains (1) are added to the substratum (brown sugar solution at 10%) and are left to stand at room temperature for fermentation (24–48 h) (2); the brown sugar solution is then fermented, forming the kefir beverage (3); finally, the kefir grains are filtered out (4), ready to start another cycle. The fermented beverage that results from step 3 is suitable for consumption (room temperature or refrigerated at 4 °C).

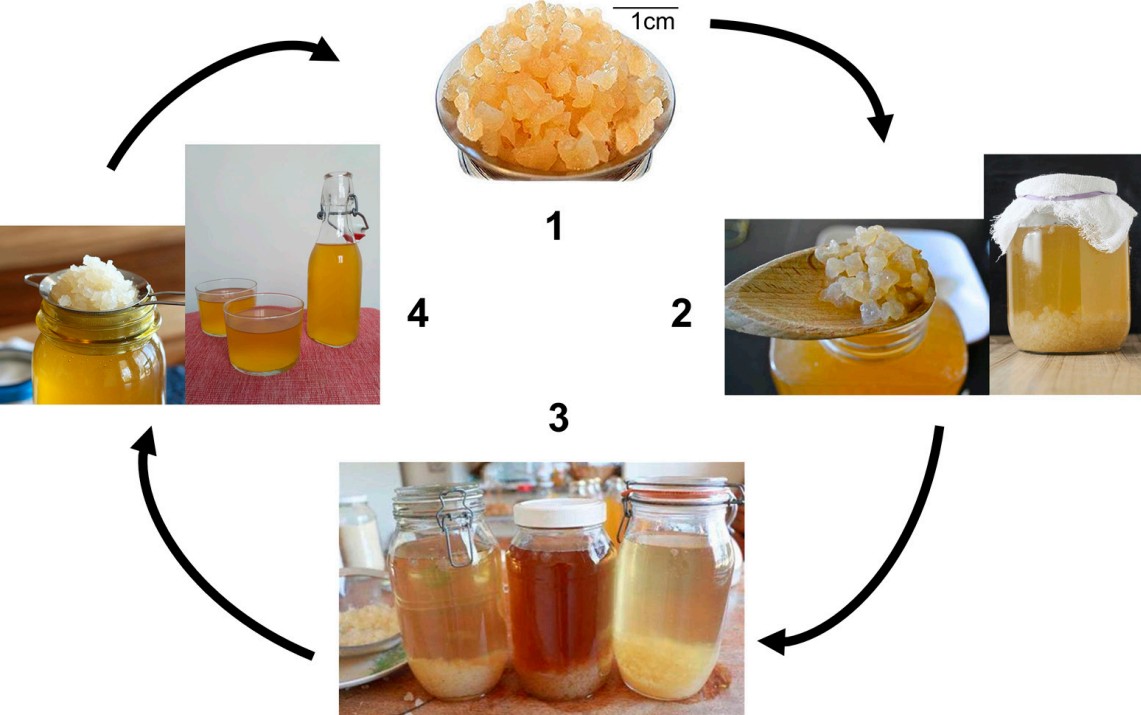

**Figure 1.** Water kefir beverage production. Water kefir grains (1), kefir grains in fermentation process (2), fermented kefir beverage (3), recovered kefir grains (4). Original and unpublished figure (authors' archive).

Kefir grains contain lactic acid bacteria (LAB), acetic acid bacteria (AAB), and yeasts [1–6]. This microbial group coexists symbiotically in the grains (kefiran), and these microorganisms are released into the fermented beverage. Water kefir grains are reused for the next fermentation, after being filtered out of the fermented beverage [1–6]. The flowchart of the production process for water kefir beverages is illustrated in Figure 1. The most commonly used source of sugar for fermentation is raw sugarcane [1–7]. The most common bacteria are *Lactobacillus*, *Lacticaseibacillus*, *Lentilactobacillus*, *Bifidobacterium*, *Oenococcus*, *Lactococcus*, *Streptococcus*, *Leuconostoc* and *Acetobacter*. The *Saccharomyces* and *Kluyveromyces* yeasts are the predominant genera [1–8]. The fermentations resulting from the metabolism of the microorganisms of kefir grains, such as alcoholic, lactic, and acetic fermentations, can generate a beverage rich in acids, such as lactic acid and acetic acid, as well as other metabolites,

such as ethanol, carbon dioxide, vitamin B12, and polysaccharides. These substances are responsible for the unique sensory characteristics of kefir [9]. Water kefir has an acidic, slightly sweet, effervescent, and slightly fermented aroma and flavor [6,9–13].

Brown sugar is the most common energy source for water kefir grains development [14–16] (Figure 1); however, the use of other carbohydrate sources may be able to support the fermentation process, resulting in tasty and healthy beverages. According to Silva et al. [17], refined and unrefined sugars can affect the microbiota of kefir differently, thus producing beverages with varying acidity and concentrations of microorganisms and metabolites. Therefore, the objective of this study was to develop kefir-based beverages with different fermentative substrates, in addition to characterizing the beverages by physicochemical and microbiological methods.

## 2. Materials and Methods

### 2.1. Raw Materials

Cane molasses sugar, coconut sugar, brown sugar, refined sugar, and demerara sugar were purchased at the local market in Salvador, Bahia, Brazil. The kefir grains were donated by the Federal University of the Recôncavo Baiano (UFRB), Cruz das Almas, BA/Brazil.

### 2.2. Substrate Preparation and Fermentation

Five different substrates were prepared for the fermentation process in triplicate. To each Erlenmeyer flask was added 350 mL deionized water and 35 g of each type of sugar (refined, coconut, demerara, brown, cane molasses). Subsequently, 35 g of water kefir grains (10%) were added to the sugary substrate. The fermentation process was carried out for a period of 48 h at a temperature of 27 °C. Samples (10 mL) were collected at times 0, 24, and 48 h of the fermentation process for physicochemical and microbiological analyses.

### 2.3. Fermentation Kinetics Analysis

The pH analysis was determined by direct potentiometry, using a digital pH meter (model K39-1014B, Kasvi, Sao Jose dos Campos, Paraná, Brazil). The determination of soluble solids was performed by digital refractometry, through the measurement of °Brix in a refractometer (model DR 201-95, Kruss, MA, USA) with a scale of 0 to 32 °Brix. For titratable acidity, 10 mL of water kefir beverage samples were pipetted into an Erlenmeyer flask, adding 50 mL of water and 3 drops of phenolphthalein solution. Titration was performed with 0.1 N sodium hydroxide solution until a pink color was reached.

The sucrose, lactic, and acetic acids were identified and quantified by high-efficiency liquid chromatography (Series 200, Perkin Elmer, Waltham, MA, USA) using a 220 mm × 4.6 mm × 10 µm polypore H column (Perkin Elmer, Waltham, MA, USA), injection volume: 10 µL, UV-Vis detector at 220 nm, flow rate: 0.8 mL/min, and mobile phase: ultrapure water acidified with $H_2SO_4$ at pH 2.0. The peaks corresponding to each acid were identified from the retention times according to the standards. A Shimadzu ion exclusion column (Shim-pack SCR-101H, 7.9 mm × 30 cm, Waltham, MA, USA) was used for carbohydrates (30 °C).

Different groups of bacteria and yeasts were enumerated using the surface-spread technique [18]. Enumeration of microorganisms was carried out in six different culture media. To characterize the total population of lactic acid bacteria (LAB), the following culture media were used for different bacterial genera: De Man, Rogosa and Sharpe agar (MRS, Oxoid, Hampshire, England) was used to enumerate *Lactobacillus*, *Lacticaseibacillus*, and *Lentilactobacillus*, M17 agar (Oxoid, Hampshire, England) to enumerate *Lactococcus*, Edwards medium (Sigma, St. Louis, MO, USA) to enumerate *Streptococcus*, and LUSM medium (Sigma, St. Louis, MO, USA) to enumerate *Leuconostoc*. Medium 254 (Sigma, St. Louis, MO, USA) was used to enumerate acetic acid bacteria (AAB), *Acetobacter* genus. All media for bacterial enumeration were supplemented with 0.4 mg/mL nystatin (Sigma, St. Louis, MO, USA). For yeast growth, Sabouraud agar medium (Sigma, St. Louis, MO, USA) supplemented with 50 mg/L of chloramphenicol (Sigma, St. Louis, MO, USA) was

used. After spreading, plates were incubated at 28 °C for 48 h for bacteria (aerobic cultivation except for MRS (anaerobic cultivation in anaerobic chamber—Anaerobe Systems, CA, USA) and 5 days for yeasts (aerobic cultivation), and colony forming units (log CFU/mL) were quantified [18].

The total bacterial growth was able to be measured because the culture media were selective, but the identification at the genus/species level was not confirmed for each culture medium. Molecular identification of lactic/acetic acid bacteria species was not made from the microbiological media, but further identification was made from the prepared kefir beverages using the PCR-DGGE technique and DNA sequencing of gel bands.

*2.4. Microbiological Identification Using PCR-DGGE—PCR-Based Denaturing Gradient Gel Electrophoresis*

The molecular technique PCR-DGGE (PCR-based denaturing gradient gel electrophoresis) was used to compare the profile of the microbial community (bacteria and yeasts) throughout the fermentation process to detect microbiological contamination. The microbiological analysis of the kefir grains and the kefir beverages (demerara, molasses, brown, refined, coconut) was carried out at the Molecular Biology Laboratory of the Federal University of Lavras—UFLA, Minas Gerais, Brazil. For the analysis, 1 g of kefir grains and 1 mL of beverage samples (0, 24, and 48 h) were used for DNA extraction [14]. The amplification was carried out in accordance with the study of Tavares et al. [14]. Each sample was centrifuged at $17,500 \times g$ for 5 min. Pellets were resuspended in 400 μL of sterilized water. Each sample was transferred into a plastic tube and was subjected to DNA extraction using a NucleoSpin Tissue kit (Macherey-Nagel, Düren, Germany). DNA extraction was performed according to the manufacturer's instructions. The genomic DNA was resuspended in sterilized water and stored at −20 °C. The bacterial community DNA was amplified with the primers 338fgc and 518r spanning the V3 region of the 16S rDNA gene [14]. The yeast community DNA was amplified using the primers NS3 and YM951r [14]. The PCR mix (25 μL) contained 0.625 U Taq DNA polymerase (Promega, Milan, Italy), 2.5 μL of buffer, 0.1 mM dNTPs, 0.2 LM of each primer, 1.5 mM $MgCl_2$, and 1 μL of DNA diluted to 10 ng/μL. The amplification was carried out as follows: template DNA was denatured for 5 min at 95 °C followed by 30 cycles of denaturing at 92 °C for 60 s, annealing at 55 °C for 60 s, and primer extension at 72 °C for 60 s. The tubes were incubated for 10 min at 72 °C for the final extension. Aliquots (2 μL) of the amplification products were analyzed by electrophoresis on 1% agarose gels before they were subjected to PCR-DGGE. The PCR products were analyzed by PCR-DGGE using a Bio-Rad DCode Universal Mutation Detection System (Bio-Rad, Richmond, CA, USA). Samples were applied to 8% ($w/v$) polyacrylamide gels in 0.5 TAE. Optimal separation was achieved with a 15–55% urea–formamide denaturing gradient for the bacterial community and a 12–50% gradient for the yeast community, where 100% is defined as 7 M urea and 40% ($v/v$) formamide. Electrophoresis was carried out for 3 h at 200 V at 60 °C, and the gels were stained with SYBR-Green (Molecular Probes, Eugene, OR, USA) (1:10,000 $v/v$) for 30 min. The gels were visualized via UV transillumination, and images were captured using a Polaroid camera (Concord, MA, USA).

The bands were excised with a sterile surgical blade and stored at −20 °C until further analysis. For the identification and analysis, the PCR-DGGE bands were excised from the acrylamide gels and the fragments were purified using a QIAEXÒ II gel extraction kit (Qiagen, Chatsworth, CA, USA). DNA recovered from each DGGE band was reamplified using the primers 338 f (without GC clamp) and 518 r for bacteria and NS3 (without GC clamp) and YM951 r for yeast. The PCR amplicons were then sequenced (Applied Biosystems, Foster City, CA, USA). GenBank searches (http://www.ncbi.nlm.nih.gov/BLAST/) (12 October 2022) were performed to determine the closest known relatives of the partial ribosomal DNA sequences obtained [19].

*2.5. Sensory Analysis*

The kefir beverages (demerara, molasses, brown, refined, coconut) were evaluated in a sensory test by 100 untrained testers, both males and females, 25–55 years of age (students and staff of the Federal University of Bahia, Brazil). The tasters were asked to indicate how much they liked or disliked each product on a 9-point hedonic scale (9 = like extremely; 1 = dislike extremely) according to overall acceptability. Evaluations of the appearance, color, flavor, and texture attributes were conducted.

The kefir beverages (20 mL) (24 and 48 h of fermentation process) were served at the same time, and the tasters were instructed to evaluate the five samples from left to right, respecting the order of presentation. The kefir beverages remained refrigerated at 4 °C until the moment of evaluation by the tasters.

In the free-listing task, tasters were asked to indicate the sensory attributes that best described the appearance, color, flavor, and texture of the samples, with no time limit being applied. To obtain the free-list terms, the total number of attributes mentioned by all tasters were calculated, and only the attributes mentioned by at least 5% of participants were included in data analysis using Smith's salience index (Dorothy D. Nevill and Steven J. Kruse, University of Florida, USA).

This "Kefir sensory analysis" project was approved by the Research Ethics Committee of the Nutrition School of the Federal University of Bahia, approval number 1.759.169.

*2.6. Statistics*

The results were evaluated using analysis of variance (ANOVA) (Ronald Aylmer Fisher, FRS, London, England). All analyses were performed in triplicate and Tukey's test (John Wilder Tukey, New Bedford, MA, USA) ($p < 0.05$) was applied for comparisons of means. A clustered heatmap was prepared in the ClustVis Webtool application (Metsalu, Tauno e Vilo, Jaak, Institute of Computer Science, University of Tartu, Estonia), using Euclidean clustering distance and the Ward algorithm as the clustering method.

**3. Results and Discussion**

*3.1. Fermentation Kinetics*

3.1.1. Physicochemical Parameters

Table 1 shows the results found for the parameters of total soluble solids, sucrose, pH, and acidity for the different water kefir fermentative substrates. The initial pH and carbohydrate value for each substrate is different, as they contain different types of commercial sugars. The pH/°Brix was not corrected for the purpose of evaluating the substrates in their original form. The initial pH values for the sugary solutions without the addition of kefir grains were 5.49 ± 0.02; 5.94 ± 0.02; 6.01 ± 0.01; 5.61 ± 0.01; and 5.47 ± 0.01 for the demerara, molasses, brown, refined, and coconut samples, respectively. After the kefir grains addition, these values lowered, as observed in Table 1 for a time of 0 h. For soluble solids and sucrose, in general, there was a reduction in their values over the analyzed period, which is explained by the metabolism of microorganisms, which are capable of using the sugars present in the substrate for the conversion of metabolites such as acetic and lactic acids. The greater presence of acids in the fermentation substrate can lower the pH and increase its acidity. Tavares et al. [14] reported that acid production during kefir fermentation is of great importance due to its inhibitory effect on product spoilage and on pathogenic microorganisms.

**Table 1.** Physicochemical analyses performed on water kefir beverages with different fermentative substrates.

| | pH | | | Acidity (% m/v) | | |
|---|---|---|---|---|---|---|
| | **0 h** | **24 h** | **48 h** | **0 h** | **24 h** | **48 h** |
| Demerara | 4.68 [cC] ± 0.03 | 4.05 [aB] ± 0.02 | 3.71 [bA] ± 0.02 | 0.29 [cC] ± 0.05 | 1.22 [cB] ± 0.02 | 1.95 [dA] ± 0.02 |
| Molasses | 5.05 [bC] ± 0.02 | 3.52 [cB] ± 0.03 | 3.38 [dA] ± 0.01 | 1.01 [aC] ± 0.01 | 2.87 [aB] ± 0.11 | 6.48 [aA] ± 0.02 |
| Brown | 5.26 [aC] ± 0.01 | 3.67 [bB] ± 0.03 | 3.41 [dA] ± 0.03 | 0.47 [bC] ± 0.05 | 2.56 [bB] ± 0.03 | 4.61 [cA] ± 0.02 |
| Refined | 4.75 [cC] ± 0.02 | 4.07 [aB] ± 0.01 | 3.88 [aA] ± 0.01 | 0.20 [cC] ± 0.05 | 1.21 [cB] ± 0.01 | 1.61 [eA] ± 0.01 |
| Coconut | 4.76 [cC] ± 0.01 | 3.59 [bcB] ± 0.01 | 3.58 [cA] ± 0.01 | 0.43 [bC] ± 0.05 | 2.75 [aB] ± 0.08 | 4.90 [bA] ± 0.08 |
| | **°Brix** | | | **Sucrose (g/L)** | | |
| | **0 h** | **24 h** | **48 h** | **0 h** | **24 h** | **48 h** |
| Demerara | 10.47 [bA] ± 0.11 | 10.37 [bA] ± 0.05 | 10.33 [aA] ± 0.05 | 0.54 [bA] ± 0.01 | 0.44 [bA] ± 0.01 | 0.41 [bA] ± 0.01 |
| Molasses | 8.17 [eA] ± 0.05 | 7.80 [dB] ± 0.10 | 7.38 [dC] ± 0.02 | 0.35 [eA] ± 0.05 | 0.18 [eA] ± 0.05 | 0.05 [eA] ± 0.01 |
| Brown | 10.03 [dA] ± 0.05 | 9.68 [cB] ± 0.02 | 9.47 [cB] ± 0.05 | 0.45 [dA] ± 0.05 | 0.19 [dA] ± 0.05 | 0.09 [dA] ± 0.01 |
| Refined | 10.80 [aA] ± 0.05 | 10.73 [aA] ± 0.05 | 11.10 [aA] ± 0.30 | 0.51 [aA] ± 0.05 | 0.50 [aA] ± 0.05 | 0.49 [aA] ± 0.05 |
| Coconut | 10.27 [cA] ± 0.05 | 9.77 [cB] ± 0.05 | 8.60 [cB] ± 0.10 | 0.47 [cA] ± 0.05 | 0.25 [cA] ± 0.05 | 0.12 [cA] ± 0.05 |

The results at 0 h, 24 h, and 48 h are presented as mean ± standard deviation. Equal lower-case letters at the same fermentation time indicate that there is no statistical difference for different samples. Equal capital letters in the different fermentation times indicate that there is no statistical difference for the same sample.

Table 1 also found a significant decrease in this parameter (soluble solids and sucrose) in water kefir developed using coconut sugar. Regarding the pH and acidity of the beverages, it is possible to verify an inverse relationship: as the pH decreased during fermentation, the acidity increased. Using molasses for the fermentative substrate resulted in a more intense pH reduction. Its titratable acidity showed the most significant increase, compared with the other substrates. The higher intensity of the fermentative process when molasses is used can be explained by its higher microbial metabolism [20,21], compared with the other samples [12,22]. According to Laureys and De Vuyst [23], natural sugar is the preferred substrate during water kefir fermentation, justifying the results found in the present study. A similar result was found for brown sugar (Table 1). Natural sugar fermentation is desirable for improving the antioxidant profile of the developed beverage because the bioavailability and bioaccessibility of a variety of compounds, including antioxidant compounds such as polyphenols and vitamins, are improved by the activity of a series of different enzymes after the fermentation process [12,21–23]. This fact is important for functional food/beverage characterization.

### 3.1.2. Microbiological Growth and Identification

Figure 2 indicates the total growth of lactic acid bacteria (*Lactobacillus*, *Lacticaseibacillus*, *Lentilactobacillus Lactococcus*, *Leuconostoc*, and *Streptococcus*) (LAB), acetic acid bacteria (*Acetobacter*) (AAB), and the total growth of yeasts throughout the water kefir fermentation process. The microbial growth pattern reflects the fermentation dynamics in relation to substrate utilization and metabolite production during fermentation [24]. It is noticed that the substrates developed with molasses and brown sugar obtained statistically higher microorganism counts than the other substrates. However, for bacteria counts (Figure 2a), refined and demerara sugars resulted in substrates with lower growth potential, while for yeasts (Figure 2b) the substrate with coconut sugar showed lower values. According to Lynch et al. [25], a low initial pH can be associated with a compromise in the fermentative process, as in the case of the beverages developed with demerara and refined sugar, which presented a pH of 4.68 and 4.75 after kefir grains addition, respectively, and obtained reduced values for microorganism growth counts.

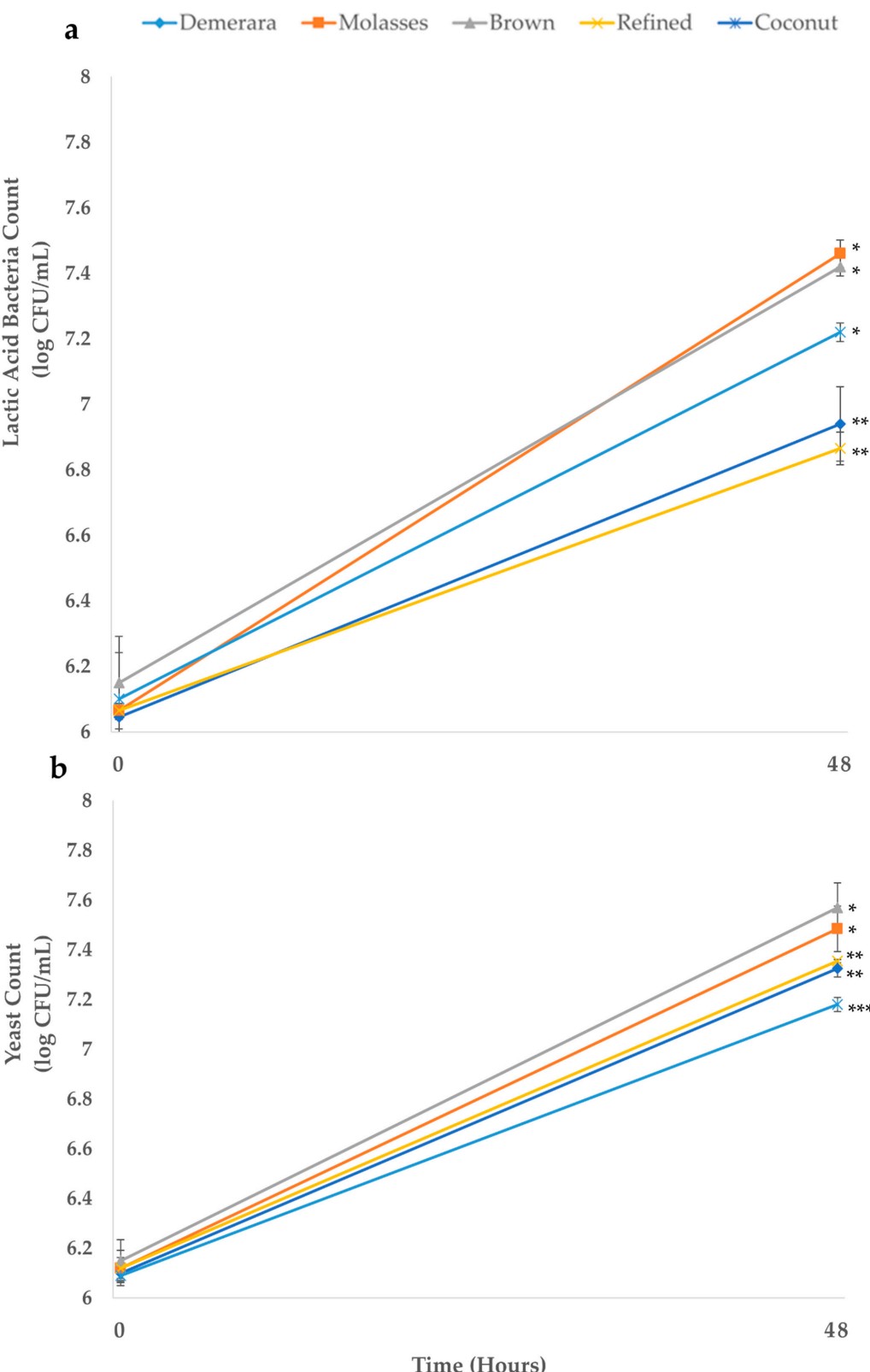

**Figure 2.** The growth of lactic/acetic acid bacteria (**a**) and yeasts (**b**) during the fermentation process of water kefir grains in different fermentative sugary substrates sugary substrates (Demerara, Molasses, Brown, Refined and Coconut). The asterisks indicate groups without a statistical difference (*, **, or ***), according to the Tukey test performed for the results at 48 h.

In a study, Tu et al. [24] verified the growth of microorganisms in water kefir to values above 7.00 log CFU/mL for LAB/AAB and yeast. In the present study, similar results were found, with only the samples produced using demerara and refined sugars not showing counts above 7.00 log CFU/mL (for LAB/AAB) after the end of the fermentative process. This indicates that the counts found, in general, would be within the recommended levels for beverages considered to be potentially probiotic [2,26].

To date, scientific research has focused on the health effects of probiotic microorganisms and their possible roles in the immune system [27]. Kefir is classed as a probiotic/prebiotic food and is popular in several countries. Prebiotic compounds such as fructooligosaccharides (FOS), galactooligosaccharides (GOS), and other oligosaccharides, as well as inulin and lactulose, were characterized in the composition of kefir grains. These compounds help to maintain the microbial symbiosis and the polysaccharide matrix (grains/kefiran), characterizing kefir as a prebiotic [14,27,28]. Its protein matrix (grains) contains various prebiotic compounds and probiotic bacteria/yeast: lactic acid bacteria, acetic acid bacteria, and yeast, coupled together with protein, casein, polysaccharides, and vitamins [14,27,28]. Kefir has specific characteristics (such as taste and aroma) that are typically attributed to the presence of a complex microbial population [14,28,29].

The increasing public demand for naturally fermented foods opens up an opportunity to construct a minimally defined microbial consortium (bacteria and yeasts) that enables the controlled and optimized production of a sensorially pleasing/functional beverage. Recent advances in "molecular biology" data integration methods have proved to be powerful in investigating the microbial contribution to metabolites production during fermentation with the aim of optimizing characteristics such as flavor production [14]. Traditional microbial identification processes are only partially selective and exclude species (bacteria/yeasts) from the total microbial population in analyzed samples [5]. Therefore, the use of molecular biology is necessary [14]. PCR-DGGE analysis (Figure 3) and DNA sequencing were used to determine the total microbial population of water kefir grains/beverages to species-level identification (Figure 4).

Kefir grains and beverages displayed the same microbial diversity during the fermentation process in the PCR-DGGE molecular method analysis (Figure 3), showing kefir beverage production to be free of microbial contamination. The microbial diversity was distributed in four bacterial genera and four yeast genera (Figure 4). Bacteria of the genera *Lactobacillus*, *Lacticaseibacillus*, *Lentilactobacillus*, *Leuconostoc*, and *Acetobacter* were found, in addition to the yeasts *Saccharomyces*, *Kluvyeromyces*, *Lachancea*, and *Kazachstania* (Figure 4).

Table 2 shows the diversity of microbial species identified by the GenBank searches using NCBI-BLAST. The identity (%) and the e-value demonstrate excellent quality in species identification in relation to the error probability (e-value $< 10 \times 10^{-100} = 99$–100 hits and e-value $< 10 \times 10^{-50} = 97$–98 hits). The genera *Lactobacillus*, *Lacticaseibacillus*, and *Lentilactobacillus* were distributed in eight different species, being the genera of dominance in the fermentation processes (Table 2). These bacteria have proven probiotic activity, mainly in their ability to aid in the body's immunity and its defense against pathogenic microorganisms [30]. Specifically, *Lentilactobacillus kefiri* produces vitamins during fermentation, and thus is able to enrich the fermented product nutritionally [29]. The lactose-fermenting yeast, *Kluyveromyces lactis* was identified in the kefir grains together with the non-lactose-fermenting yeasts *Lachancea meyersii*, *Kazachstania aerobia*, and *Saccharomyces cerevisiae*. These yeasts represent the most commonly identified yeast isolates in kefir grains [14,17,23–29]. Furthermore, in relation to yeasts, *Saccharomyces cerevisiae* has immunomodulatory potential in its ability to stimulate the specific proliferative response of T lymphocytes and also because of its antibacterial activity [31].

**a**

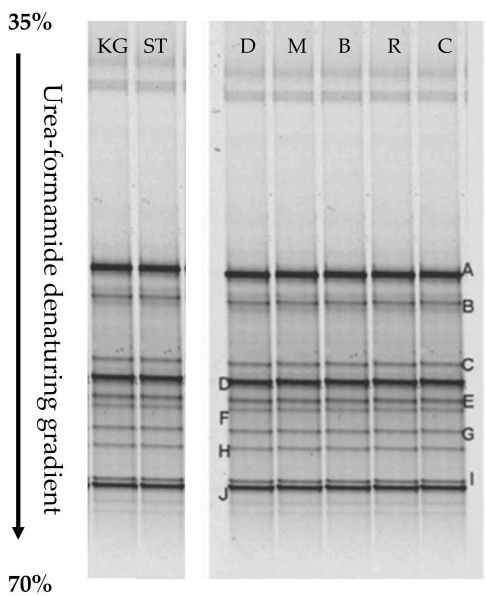

KG – Kefir grains
ST - Start time
D - Demerara
M - Molasses
B - Brown
R - Refined
C - Coconut

**b**

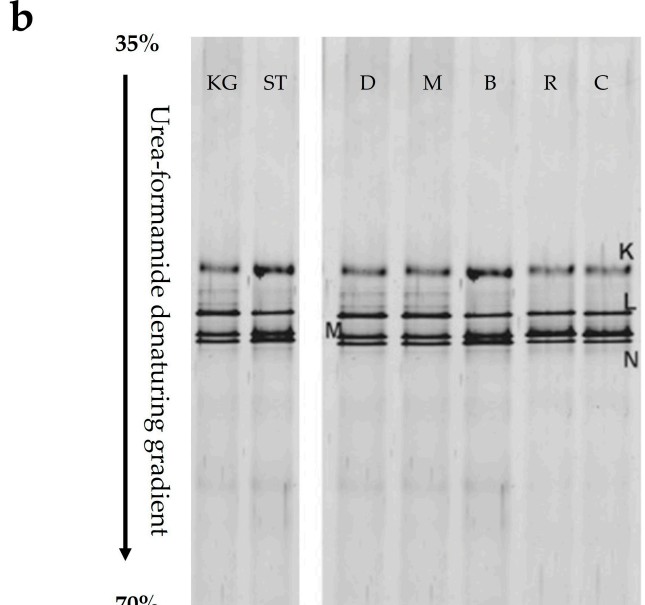

**Figure 3.** Profiles of microbial communities from water kefir grains and beverages. (**a**) Prokaryotic group (Band A: *Lacticaseibacillus paracasei* (Access number—AB368902.1), Band B: *L. kefiri* (Access number—AB3626680.1), Band C: *L. parabuchneri* (Access number—AB368914.1), Band D: *L. casei* (Access number—EU626005.1), Band E: *L. paracasei* subsp. *paracasei* (Access number—NR025880.1), Band F: *L. paracasei* subsp. *tolerans* (Access number—AB181950.1), Band G: *L. buchneri* (Access number—FJ867641.1), Band H: *Lactococcus lactis* (Access number—EU194346.1), Band I: *Leuconostoc citreum* (Access number—FJ378896.1), and Band J: *Acetobacter lovaniensis* (Access number—AB308060.1)). (**b**) Eukaryotic group (Band K: *Kluyveromyces lactis* (Access number—AJ229069.1), Band L: *Saccharomyces cerevisiae* (Access number—EU649673.1), Band M: *Kazachstania aerobia* (Access number—AY582126.1), and Band N: *Lachancea meyersii* (Access number—AY645661.1)).

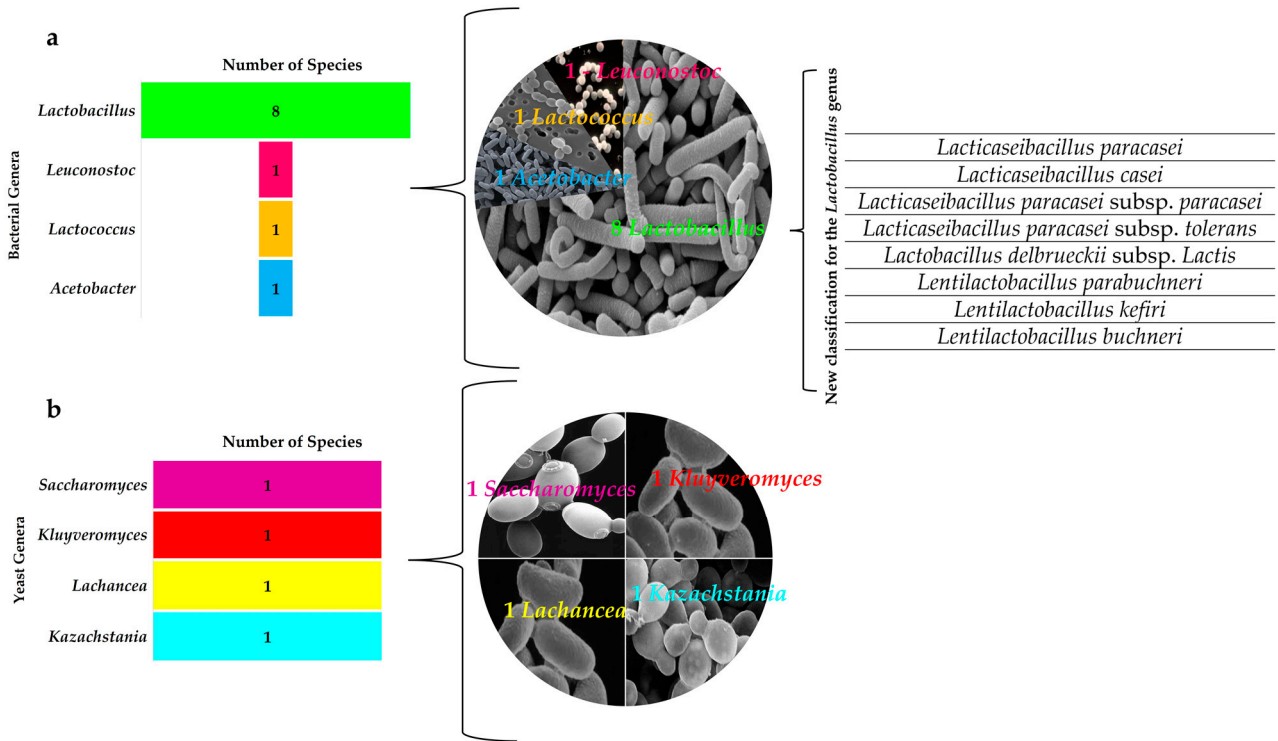

**Figure 4.** Diversity of the microbial genera present in water kefir grains and beverages. (**a**) Bacteria genera; (**b**) Yeast genera.

**Table 2.** Microorganisms present in water kefir grains and beverages (demerara, molasses, brown, refined, coconut) as identified by PCR-DGGE analysis and species sequencing.

| Microbial Species (New Name) | Microbial Species (Current Name) | NCBI-BLAST Accession Number Access date: 12 October 2022 | Identity (%) | e-Value |
|---|---|---|---|---|
| *Lacticaseibacillus paracasei* | *Lactobacillus paracasei* | AB368902.1 | 99 | $<10 \times 10^{-100}$ |
| *Lacticaseibacillus casei* | *Lactobacillus casei* | EU626005.1 | 98 | $<10 \times 10^{-50}$ |
| *Lacticaseibacillus paracasei* subsp. *paracasei* | *Lactobacillus paracasei* subsp. *paracasei* | NR025880.1 | 98 | $<10 \times 10^{-50}$ |
| *Lacticaseibacillus paracasei* subsp. *tolerans* | *Lactobacillus paracasei* subsp. *tolerans* | AB181950.1 | 99 | $<10 \times 10^{-100}$ |
| *Lactobacillus delbrueckii* subsp. *lactis* | *Lactobacillus lactis* | EU194346.1 | 98 | $<10 \times 10^{-50}$ |
| *Lentilactobacillus parabuchneri* | *Lactobacillus parabuchneri* | AB368914.1 | 99 | $<10 \times 10^{-100}$ |
| *Lentilactobacillus kefiri* | *Lactobacillus kefiri* | AB3626680.1 | 99 | $<10 \times 10^{-100}$ |
| *Lactococcus lactis* | *Lactococcus lactis* | EU194346.1 | 99 | $<10 \times 10^{-100}$ |
| *Leuconostoc citreum* | *Leuconostoc citreum* | FJ378896.1 | 99 | $<10 \times 10^{-100}$ |
| *Lentilactobacillus buchneri* | *Lactobacillus buchneri* | FJ867641.1 | 99 | $<10 \times 10^{-100}$ |
| *Acetobacter lovaniensis* | *Acetobacter lovaniensis* | AB308060.1 | 99 | $<10 \times 10^{-100}$ |
| *Saccharomyces cerevisiae* | *Saccharomyces cerevisiae* | EU649673.1 | 99 | $<10 \times 10^{-100}$ |

**Table 2.** *Cont.*

| Microbial Species (New Name) | Microbial Species (Current Name) | NCBI-BLAST Accession Number Access date: 12 October 2022 | Identity (%) | e-Value |
|---|---|---|---|---|
| *Kluyveromyces lactis* | *Kluyveromyces lactis* | AJ229069.1 | 99 | $<10 \times 10^{-100}$ |
| *Lachancea meyersii* | *Lachancea meyersii* | AY645661.1 | 99 | $<10 \times 10^{-100}$ |
| *Kazachstania aerobia* | *Kazachstania aerobia* | AY582126.1 | 99 | $<10 \times 10^{-100}$ |

The e-value indicates the number of alignments that would be expected to show score values equal to or better than the one found by chance, given the size of the database. e-value $< 10 \times 10^{-100}$ = 99–100 hits. e-value $< 10 \times 10^{-100}$ = 97–98 hits.

Consumer interest in functional food is evident, as the global market for this product is steadily growing [32]. This has led to the incorporation of probiotic microorganisms into food/beverages to result in functional food. Some studies have demonstrated that it is possible to produce foods by combining different ratios of probiotic bacteria/yeasts. This factor may serve as an excellent option for a probiotic/healthy diet [28,29]. To develop the probiotic potential of kefir beverages, the bacteria/yeast population needs to remain viable in the final product [29]. This was demonstrated in this study (Figure 2). Bacteria (LAB/AAB) and yeasts from the kefir grains remained viable in the fermented beverages containing different types of sugars produced in this study. The diverse microbiological profile of water kefir grains/beverages has been attributed to the various geographic world regions from which it originated. Variations in the microbial population from the same kefir grain have recently been demonstrated by massive sequencing studies on water kefir beverages obtained through successive fermentations in different substrates [1,3,6,29]. However, in this study the microbial biodiversity/ecology of bacteria and yeasts was constant during the fermentation process for all types of sugary substrates used.

3.1.3. Organic Acids Production

Figure 5 shows the production of organic acids by the microorganisms in water kefir during the fermentation process. Corroborating the results found in Figure 2, a higher number of microorganisms was verified in the substrates fermented with molasses and brown sugar. It is also possible to verify in these fermentations a higher presence of lactic and acetic acids. According to Viana et al. [33], organic acids are metabolites produced during the fermentation of kefir grains. The presence of these organic acids in kefir beverages has important functions, such as the inhibition of pathogen proliferation, in addition to the characteristic flavor of fermented beverages [33].

The values for lactic acid were close to those presented by Destro et al. [34], namely 0.78 g/L for kefir prepared with brown sugar and jaboticaba. On the other hand, Magalhães et al. [35] verified higher values of acetic acid at the end of the fermentation of traditional Brazilian water kefir, which may explain the lower intensity of metabolism of acetic acid bacteria in the fermentative process in the present study.

According to Puerari et al. [36], a higher concentration of organic acids may be a factor that compromises the sensory acceptance of kefir beverage, as it has a sweeter flavor when it has low acidity values. In the present study, the beverages developed with refined and demerara sugars may have greater acceptance by the consumer market due to their lower acid values.

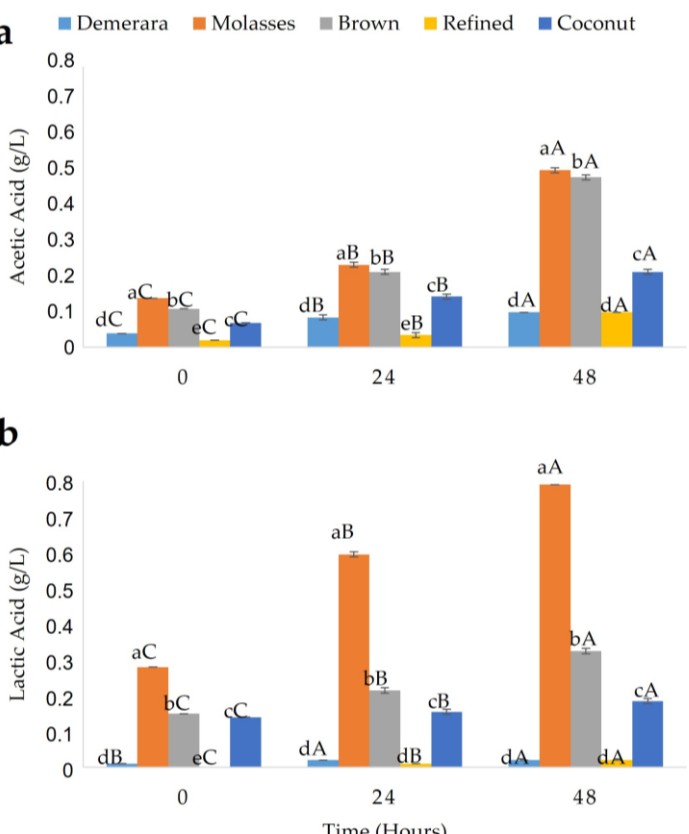

**Figure 5.** Behavior of acetic (**a**) and lactic (**b**) acids during the fermentation process of water kefir grains in different fermentation sugary substrates (Demerara, Molasses, Brown, Refined and Coconut). The same lower-case letters at the same fermentation time indicate that there is no statistical difference for the different samples. Equal capital letters at different fermentation times indicate that there is no statistical difference for the same sample, according to the Tukey test.

Worldwide consumption of water kefir beverages has increased during recent years [1–6,23,25,27–29,33–36]. Despite homemade water kefir beverage being the product most frequently consumed, many industrialized commercial water kefir beverages can be found in some markets in different countries. Sometimes it is sold as an "artisanal kefir beverage" which is not regulated by food legislation. Nevertheless, trends over recent decades, mainly related to regulations on probiotic strains, have also begun to affect water kefir grains/beverages, thus requiring that these products comply with "food legislation" in the different countries in which they are consumed.

### 3.1.4. Kefir Grains Growth

Regarding the biomass growth of the water kefir grains, Figure 6 shows the development of the water kefir grains for the different fermentative substrates. Statistically, water kefir beverages fermented with molasses, coconut, and brown sugars showed a significant increase in the biomass of the water kefir grains. However, demerara and refined sugars stimulated lower biomass growth of the water kefir grains as measured at the end of the fermentative process. The water kefir grains fermented in the medium containing molasses showed the highest percentage of biomass growth (19.15%) in 48 h of fermentation.

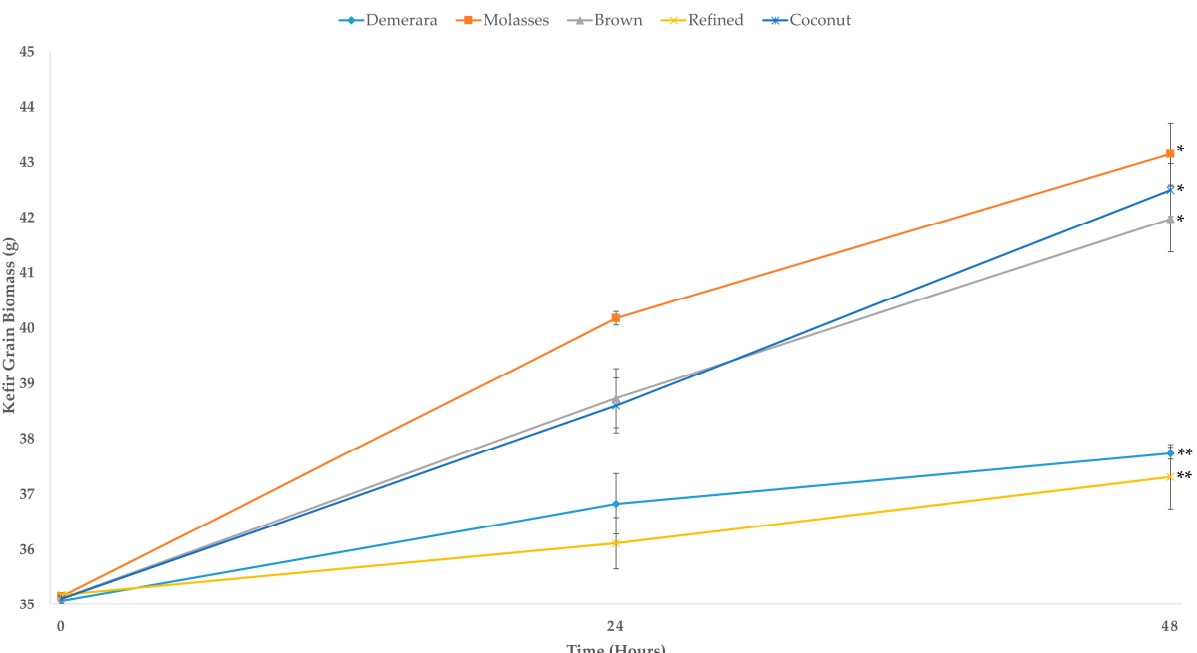

**Figure 6.** Biomass growth of water kefir grains during the fermentation process in different sugary substrates (Demerara, Molasses, Brown, Refined and Coconut). The asterisks indicate groups without a statistical difference (* or **) according to the Tukey test performed for the results at 48 h.

Therefore, as regards the higher biomass production of water kefir grains, this substrate (molasses) obtained better results. The water kefir grains fermented in the refined sugar medium showed the lowest yield of biomass production (5.71%) in relation to its initial weight. A possible explanation for the low biomass production could be that the microorganisms in water kefir present a compromised metabolism in the presence of refined sugars. Several mechanical and chemical processes are carried out during the industrial processing of refined sugars. As a result, processed carbohydrates may not be metabolized by the microorganisms in water kefir grains, or may have inhibitory characteristics, such as antimicrobial effect [37,38].

### 3.2. Multivariate Clustering Analysis

The clustering of the multivariate statistical analysis in the form of a heatmap is presented in Figure 7. The analysis was performed considering the physicochemical and microbiological parameters of the fermented water kefir beverages according to the type of sugar used and the fermentation time. Two larger clusters resulted from the analyses. The first cluster, on the left, is subdivided into Brix, lactic-acid bacteria, and yeast.

The cluster on the right is subdivided into acidity, lactic/acetic acid, and pH. Water kefir beverages display a high degree of subdivision. There is an initial division into two larger clusters. The upper cluster presents the samples containing coconut sugar at 24 h, brown sugar at 24 h and molasses at 48 h. These water kefir beverages showed similar characteristics, with high lactic acid bacteria counts and more intense fermentation. However, beverages containing refined and demerara sugars also formed clusters among themselves, indicating that they are similar because of the less noticeable fermentation process, in addition to lower counts of microorganisms—making them less suitable for the development of water kefir beverages.

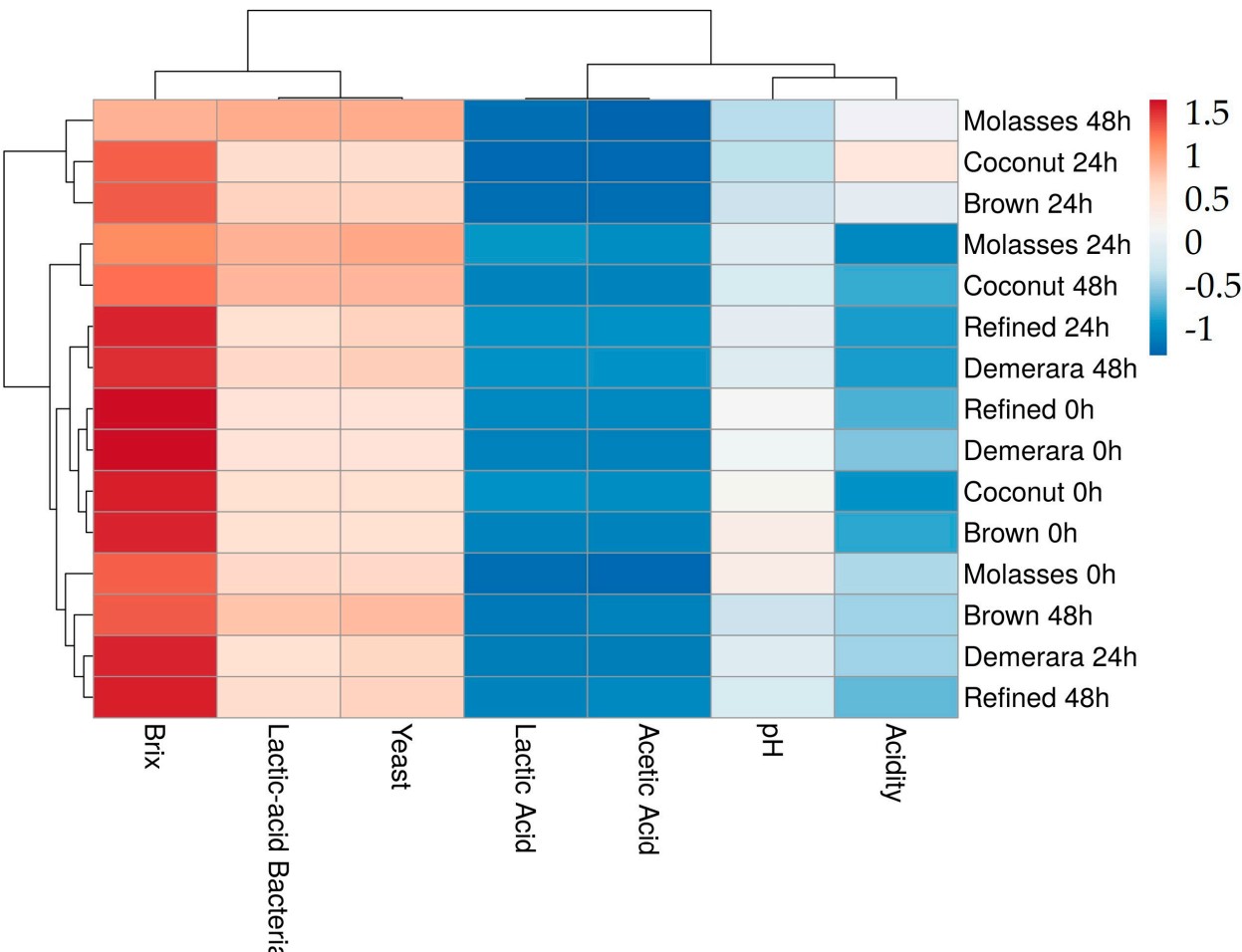

**Figure 7.** Multivariate analysis in the form of a Heatmap of the physicochemical and microbiological parameters of water kefir beverages fermented from different sugary substrates (Demerara, Molasses, Brown, Refined and Coconut).

*3.3. Sensory Analysis*

Of the tasters who participated in the study, 77% were regular kefir consumers (Supplementary Material—Figure S1. Percentage of respondents who know kefir). Among the respondents, 42% had already consumed kefir (monthly/rarely), while 58% had never consumed kefir. Most consumers stated that they consume kefir for its health benefits (32%), followed by eating trends. Several authors believe that the increase in kefir consumption is linked to the recognition of its benefits to human health [2,4,27–29,33–40].

The results of the sensory acceptance testing of the water kefir beverages are presented in Figure 8. There was no sensory difference between kefir beverages after 24 and 48 h of fermentation. Testers scored both kefir beverages as having the same characteristics. The sensory attributes of the kefir beverages scored between 7 and 8.1 on a 9-point hedonic scale, indicating "like moderately" to "like extremely", depending on the product and the sensory attribute. Differences ($p < 0.05$) between the kefir beverages were observed in respect of color and flavor. For global acceptance, the mean scores ranged ($p < 0.05$) from 7.7 to 8.4 for the kefir beverages: demerara (7.7), coconut (7.8), refined (8.0), molasses (8.0), and brown (8.4). Smith's salience index indicates that the brown-sugar kefir beverage was better accepted among the tasters. This fact can be explained by the difference in metabolite compositions that intensify the taste of the beverage, and further by the intense microbial growth and, consequently, a better fermentation process.

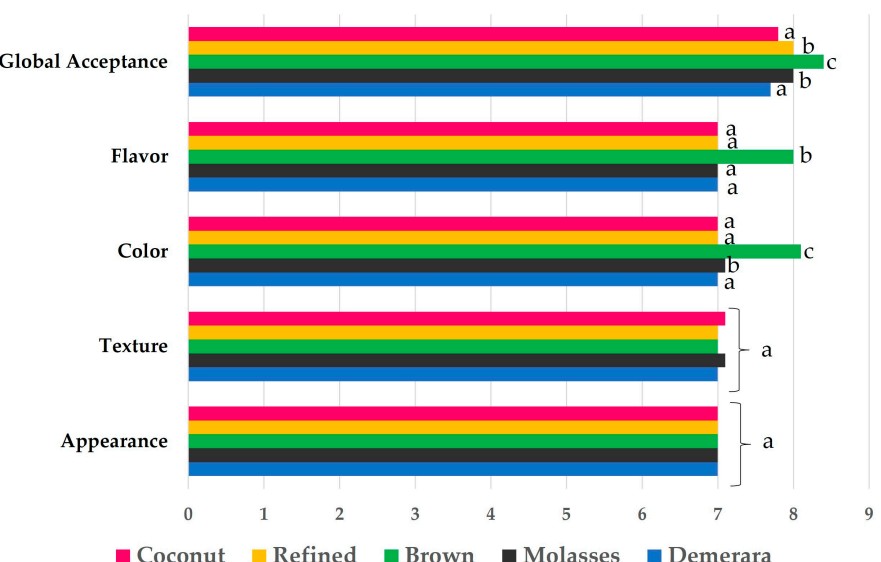

**Figure 8.** Water kefir beverages sensory evaluation. Different letters for each attribute indicate that there is a statistical difference according to the Tukey test.

## 4. Conclusions

In general terms, the sugars evaluated in the present study resulted in good substrates for the development of new water kefir beverages. The lactic/acetic acid bacteria and yeasts were able to grow in different substrates, generating beverages with probiotic potential and nutritional characteristics [41–45]. The substrates containing cane molasses and brown sugar showed more intense fermentative processes, verified through higher microorganism counts and the production of organic acids. This can be explained by the fact that they are sugars with more interesting intrinsic characteristics, such as less refinement in industrial processing. On the other hand, substrates containing refined and demerara sugars caused a more discreet fermentation. Statistically, the brown-sugar kefir beverage was better accepted among the tasters.

Developing a functional understanding of the microbiota is essential for a consistent commercial-scale water kefir product. Our results highlight the relatively constant microbial ecology evolution of water kefir through the fermentation process. These results may provide a reproducible industrial production process for water kefir beverages. Finally, greater understanding of these microbial relationships will also facilitate the construction of defined microbial strain consortia that could reproduce the key features of water kefir.

Further studies are needed to evaluate the influence of these new water kefir beverages with probiotic and functional properties.

**Supplementary Materials:** The following supporting information can be downloaded at: https://www.mdpi.com/article/10.3390/fermentation9040384/s1, Figure S1. Percentage of respondents who know kefir.

**Author Contributions:** Conceptualization, C.T.P.M., P.P.L.G.T., R.Q.N., K.T.M.-G. and M.E.d.O.M.; data curation, P.P.L.G.T., C.O.d.S., K.T.M.-G. and M.E.d.O.M.; formal analysis, C.T.P.M., P.P.L.G.T., R.Q.N. and E.A.d.A.; funding acquisition, K.T.M.-G. and M.E.d.O.M.; investigation, C.T.P.M., P.P.L.G.T. and K.T.M.-G.; methodology, P.P.L.G.T., K.T.M.-G. and M.E.d.O.M.; project administration, M.E.d.O.M.; resources, K.T.M.-G., R.C.d.C.A. and M.E.d.O.M.; program, P.P.L.G.T. and K.T.M.-G.; supervision, K.T.M.-G. and M.E.d.O.M.; validation, P.P.L.G.T., K.T.M.-G., R.C.d.C.A. and M.E.d.O.M.; writing—original draft preparation, P.P.L.G.T., K.T.M.-G. and M.E.d.O.M.; writing—original draft, C.T.P.M.; writing—review and editing, P.P.L.G.T., K.T.M.-G. and M.E.d.O.M. All authors have read and agreed to the published version of the manuscript.

**Funding:** The authors are grateful to the Coordenação de Aperfeiçoamento de Pessoal de Nível Superior—Brazil (Capes)—Code 001. Process No: 88887.682738/2022-00—UFBA.

**Institutional Review Board Statement:** Not applicable.

**Informed Consent Statement:** Not applicable.

**Data Availability Statement:** Data available on request.

**Acknowledgments:** The authors would like to dedicate this article to Janice Izabel Druzian (*in memoriam*), a professor at the Federal University of Bahia who was always available to share her immeasurable knowledge. The authors thank the following Brazilian agencies: the Conselho Nacional de Desenvolvimento Científico e Tecnológico (CNPq), and the Coordenação de Aperfeiçoamento de Pessoal de Nível Superior (CAPES).

**Conflicts of Interest:** The authors declare no conflict of interest.

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
