# Peer review of "Non-Conventional Sucrose-Based Substrates: Development of Non-Dairy Kefir Beverages with Probiotic Potential"

_fermentation, doi:10.3390/fermentation9040384_

Round 1

Reviewer 1 Report (Previous Reviewer 2)

The revised manuscript has been improved.

Author Response

Dear Reviewer

The authors would like to thank all the suggestions previously described by the reviewer for the previous version submitted to Journal Fermentation. We have improved the manuscript and the English language writing. We hope that this version will be accepted by the reviewer for publication in Fermentation.

Reviewer 2 Report (New Reviewer)

The topic of the work is interesting. The following suggestions can be used to improve the manuscript:

-Fig 1 should stands after L68. The description of Fig 1 could be presented in less detail for better clarity and transparency.

-L104 Federal University of the Recôncavo Baiano

Add city and country.

-L185 Is the sensor analysis done with fresh samples or samples stored for 24h, 48h? This needs to be specified.

-L270 Kefir belongs to the probiotic/prebiotic food

Why is kefir characterized as a prebiotic food?  

-The methods are written in the style of a laboratory journal. It is commendable that the authors wrote down all the details, but it is necessary to shorten the parts that are implied. (For example L 128- 135 Different groups of bacteria and yeasts were enumerated by the surface spread technique., plating in triplicate 100 µL of each diluted sample (10-1 to 10-6 ). Enumeration of microorganisms was carried in six different culture media. To characterize the total population of lactic acid bacteria (LAB) the following culture media were used for different bacterial genera: De Man, Rogosa and Sharpe Agar (MRS, Oxoid, Hampshire, England) was used to enumerate Lactobacillus, M17 agar (Oxoid, Hampshire, England) to enumerate Lactococcus, Edwards medium (Sigma, St. Louis, USA) to enumerate Streptococcus, and LUSM medium (Sigma, St. Louis, USA) to enumerate Leuconostoc…)

...also, please add whether the conditions during incubation were microaerophilic or aerobic?

The results section shows the total growth of LAB, not the growth obtained on the media listed in the Methods (MRS, M17, LUSM...). It is not clear what the authors did and what they presented as a result. Align Methods and Results.

-Authors should consider to present Fig 8. (L433) in Supplementary file. It would be easier to read the paper if some details were transferred to a supplementary file.

Author Response

The authors are grateful for the Reviewer's suggestions as they considerably improved the manuscript. We performed all suggested corrections. Corrections are highlighted in green in the attached manuscript.

The topic of the work is interesting.

Answer: We are grateful to the reviewer for recognizing our work.

The following suggestions can be used to improve the manuscript:

-Fig 1 should stands after L68. 

Answer: Corrected in the manuscript.

The description of Fig 1 could be presented in less detail for better clarity and transparency.

Answer: Figure 1 caption has been corrected and the full description of the figure has been added in the manuscript text.

-L104 Federal University of the Recôncavo Baiano

Add city and country.

Answer: This information has been added to the manuscript.

-L185 Is the sensor analysis done with fresh samples or samples stored for 24h, 48h? This needs to be specified.

Answer: The sensory was performed on kefir beverages after 24 and 48 hours of fermentation process. The kefir beverages remained refrigerated at 4ºC until the moment of evaluation by the tasters. There was no sensory difference between kefir beverages after 24 and 48 hours of fermentation. Testers scored both kefir beverages as having the same characteristics. This has been added in the text of the manuscript.

-L270 Kefir belongs to the probiotic/prebiotic food

Why is kefir characterized as a prebiotic food? 

Answer: Prebiotic compounds such as fructooligosaccharides (FOS), galactooligosaccharides (GOS) and other oligosaccharides, inulin and lactulose were characterized in the composition of kefir grains. These compounds help to maintain the microbial symbiosis and the polysaccharide matrix (grains/kefiran), characterizing kefir as a prebiotic. This has been added in the text of the manuscript.

-The methods are written in the style of a laboratory journal. It is commendable that the authors wrote down all the details, but it is necessary to shorten the parts that are implied. (For example L 128- 135 Different groups of bacteria and yeasts were enumerated by the surface spread technique., plating in triplicate 100 µL of each diluted sample (10-1 to 10-6 ). Enumeration of microorganisms was carried in six different culture media. To characterize the total population of lactic acid bacteria (LAB) the following culture media were used for different bacterial genera: De Man, Rogosa and Sharpe Agar (MRS, Oxoid, Hampshire, England) was used to enumerate Lactobacillus, M17 agar (Oxoid, Hampshire, England) to enumerate Lactococcus, Edwards medium (Sigma, St. Louis, USA) to enumerate Streptococcus, and LUSM medium (Sigma, St. Louis, USA) to enumerate Leuconostoc…)

...also, please add whether the conditions during incubation were microaerophilic or aerobic?

Answer: A reference was cited for the cultivation method, and this was summarized in the manuscript. The aerobic and anaerobic cultivation method has been added in the text.

The results section shows the total growth of LAB, not the growth obtained on the media listed in the Methods (MRS, M17, LUSM...). It is not clear what the authors did and what they presented as a result. Align Methods and Results.

Answer:  Lines 251 to 253 report that the sum total of bacterial growth in different culture media was performed. The sum total of yeast growth was also reported. The sum total of bacterial growth was performed because the culture media are selective but the identification at the genus/species level was not confirmed for each culture medium.

-Authors should consider to present Fig 8. (L433) in Supplementary file. It would be easier to read the paper if some details were transferred to a supplementary file.

Answer: Figure 8 has been added to the Supplementary Material - Figure S1. Figure 9 has been corrected to Figure 8 in the manuscript.

Reviewer 3 Report (New Reviewer)

There is a growing number of people who are interested in vegan products. It is important that the consumer has a choice whether he wants to consume fermented milk or vegan products, including non-dairy kefir beverages.

Therefore, the topic of the research is fully justified. The main objectives, the description of the methods and the presentation of the results are presented. However, I have a few comments that improve the quality of the manuscript:

1. Abstract is too long - please shorten it to the most important results.

2. In Line 23: I think that "milk allergies, non-tolerant to lactose" is not a reason for "increasing interest in the consumption of non-dairy fermented foods/beverages". Please remove these statements because they do not appear in the rest of the manuscript and there is no literature listed.

3. In Line 109: please specify the specific strains that you used in "kefir grains" and indicate from the specification what is the optimum temperature range for development.

4. I have doubts about the pH value - are you sure that the pH values in 0 days are correct? After adding all substrates, the pH should be close to that of deionized water. Please provide the pH of the zero sample without water kefir grains.

5. Please correct the name of Lactobacillus throughout the manuscript (according to https://isappscience.org/new-names-for-important-probiotic-lactobacillus-species/ )

6. In Figure 2: Fermentation lasted 48 hours (according to information in Line 110) - please correct the graph (the line is from ... to ..., i.e. closed. Fermentation has been interrupted. It should be a line segment, not a half line).

7. Conclusion is too long - please shorten it. Also the statement "Our results indicated that genera of bacteria (e.g., Lactobacillus, Lactococcus, Leuconostoc and Acetobacter) and yeast (e.g., Saccharomyces, Kluyveromyces, Lachancea, and Kazakhstan) were the bacteria/yeast present in water kefir grain beverages" is obvious, because “The kefir grains were donated by the Federal University of the Recôncavo Baiano.”

Author Response

Comments and Suggestions for Authors

The authors are grateful for all requests from the reviewer.

There is a growing number of people who are interested in vegan products. It is important that the consumer has a choice whether he wants to consume fermented milk or vegan products, including non-dairy kefir beverages.

Therefore, the topic of the research is fully justified. The main objectives, the description of the methods and the presentation of the results are presented. However, I have a few comments that improve the quality of the manuscript:

Request 1. Abstract is too long - please shorten it to the most important results.

Answer: Thank you for the suggestion. The abstract, indeed, was too long with a word count of 351. It was narrowed down to 225, focusing on the most important information regarding our study.

Request 2. In Line 23: I think that "milk allergies, non-tolerant to lactose" is not a reason for "increasing interest in the consumption of non-dairy fermented foods/beverages". Please remove these statements because they do not appear in the rest of the manuscript and there is no literature listed.

Answer: These statements were removed as suggested.

Request 3. In Line 109: please specify the specific strains that you used in "kefir grains" and indicate from the specification what is the optimum temperature range for development.

Answer: Thank you for the suggestion. No specific strains were selected, as we used the entirety of the kefir grains, therefore, a plethora of symbiotic microorganisms.

Request 4. I have doubts about the pH value - are you sure that the pH values in 0 days are correct? After adding all substrates, the pH should be close to that of deionized water. Please provide the pH of the zero sample without water kefir grains.

Answer: We appreciated this questioning. As we reviewed our results, the pH values were found to be incorrect and were then changed to the correct results verified in our research. Discussions regarding the pH were updated, considering this (including the Figure 7).

Request 5. Please correct the name of Lactobacillus throughout the manuscript (according to https://isappscience.org/new-names-for-important-probiotic-lactobacillus-species/ )

Answer: Thank you for sharing this new knowledge regarding the change in some former Lactobacillus bacteria’s genera. We updated this throughout the manuscript, as suggested (indicated by the teal color).

Request 6. In Figure 2: Fermentation lasted 48 hours (according to information in Line 110) - please correct the graph (the line is from ... to ..., i.e. closed. Fermentation has been interrupted. It should be a line segment, not a half line).

Answer: Thank you for the suggestion. Figure 2 was updated in order to fill the suggestion. It is now a line segment.

Request 7. Conclusion is too long - please shorten it. Also the statement "Our results indicated that genera of bacteria (e.g., Lactobacillus, Lactococcus, Leuconostoc and Acetobacter) and yeast (e.g., Saccharomyces, Kluyveromyces, Lachancea, and Kazakhstan) were the bacteria/yeast present in water kefir grain beverages" is obvious, because “The kefir grains were donated by the Federal University of the Recôncavo Baiano.”

Answer: The conclusion was shortened, focusing more on the most important findings of our research. Also, the statement indicated was removed, considering its obviousness.

Round 2

Reviewer 2 Report (New Reviewer)

The manuscript has been significantly improved. There is only one query that needs more attention:

L 129-132: Different groups of bacteria and yeasts were enumerated by the surface spread technique, [18]. Enumeration of microorganisms was carried in six different culture media. To characterize the total population of lactic acid bacteria (LAB) the following culture media were used for different bacterial genera…

If the author intended to represent only the total LAB population in the manuscript, they should: provide CFU/mL values for each group of bacteria in an Supplementary file (if the authors stated that the enumeration of microorganisms was carried in six different culture media, readers will expect to be able to find CFU/mL for each microbe), or to explain in the methods why they chose to report the total number of LAB rather than CFU/mL from the various selected media (so please enter the provided answer in the manuscript: The sum total of bacterial growth was performed because the culture media are selective but the identification at the genus/species level was not confirmed for each culture medium… molecular identification of LAB species was not investigated from microbiological media, but further identification was made from prepared beverages…)

Author Response

Comments and Suggestions for Authors

The authors are grateful for the reviewer's request.

The manuscript has been significantly improved. There is only one query that needs more attention:

L 129-132: Different groups of bacteria and yeasts were enumerated by the surface spread technique, [18]. Enumeration of microorganisms was carried in six different culture media. To characterize the total population of lactic acid bacteria (LAB) the following culture media were used for different bacterial genera…

If the author intended to represent only the total LAB population in the manuscript, they should: provide CFU/mL values for each group of bacteria in an Supplementary file (if the authors stated that the enumeration of microorganisms was carried in six different culture media, readers will expect to be able to find CFU/mL for each microbe), or to explain in the methods why they chose to report the total number of LAB rather than CFU/mL from the various selected media (so please enter the provided answer in the manuscript: The sum total of bacterial growth was performed because the culture media are selective but the identification at the genus/species level was not confirmed for each culture medium… molecular identification of LAB species was not investigated from microbiological media, but further identification was made from prepared beverages…)

Answer: This has been added in the manuscript methodology (2.3. Fermentation Kinetics Analysis) as requested by the reviewer:

“The sum total of bacterial growth was performed because the culture media are selective but the identification at the genus/species level was not confirmed for each culture medium. Molecular identification of lactic/acetic acid bacteria species was not investigated from microbiological media, but further identification was made from prepared kefir beverages by PCR-DGGE technique and DNA sequencing of gel bands.”

Reviewer 3 Report (New Reviewer)

Thanks for the revised manuscript. Accepts Authors' responses, but I have only one request (valuable remark): Please just look at Figure 2, Figure 6 - time cannot be negative. Please start the x-axis (abscissa) from 0 and not from negative values. It will be legible - please see other manuscipts (concerning the graphical presentation of fermentation kinetics) - time 0 starts right on the ordinate (y) axis (the intersection with the ordinate axis).

Author Response

Comments and Suggestions for Authors

Thanks for the revised manuscript. Accepts Authors' responses, but I have only one request (valuable remark): Please just look at Figure 2, Figure 6 - time cannot be negative. Please start the x-axis (abscissa) from 0 and not from negative values. It will be legible - please see other manuscipts (concerning the graphical presentation of fermentation kinetics) - time 0 starts right on the ordinate (y) axis (the intersection with the ordinate axis).

Response:

Dear Reviewer,

We appreciate your requests. We have corrected the two Figures (2 and 6) as requested and these corrected Figures are in the attached manuscript.

This manuscript is a resubmission of an earlier submission. The following is a list of the peer review reports and author responses from that submission.

Round 1

Reviewer 1 Report

The authors likely mixed up water kefir and milk kefir  with respect to microbiology, substrates, fermentation and product characteristics. The authors made major references to water kefir but used milk kefir grains. The starting pH of the fermentation substrates differed substantially and would have compromised the findings for fair comparisons. In addition, it seemed that only single fermentation trial (lack of biological replicates in triplicate) was done. Triplicate (technical) analyses only showed analytical reproducibility. Scientific writing is weak, incoherent, lacking insightful discussions.

Author Response

Response:

Dear Reviewer

The authors welcome your comments and contribution;

We used water kefir grains for this study. Figure 1 was added to the manuscript for this demonstration;

Changed and updated references to water kefir;

The initial pH was different because different substrates were used. The manuscript discussion has been improved;

All fermentation processes were performed in triplicate. We corrected the methodology.

The manuscript has been corrected in terms of English wording, results, figures, discussion, and conclusion.

date of submission

19 January 2023

Date of this review

24 Jan 2023 12:32:45

Reviewer 2 Report

The article discussess a very interesting topic. The experimental plan is well organized. After a little review, the work deverses to be published. 

Line 85: For bacterial count, the MRS........What does bacterial count mean?indicate for which microrganisms MRS medium was used.  Did the incubation take place in anaerobiosis? If not why?

Line 87: Explain the sampling method. As described may be difficult for readers to understand and difficult to replicate. 

Line 122: Where is table 20?

Line 150: The figure 1 shows the cell concentration of lactic acid bacteria and yeast....I reccomend editing the caption

Line 156: count above 7.00... add CFU/ml

Line 213: Figure 2a  error bars are missing in the graph 

Author Response

Comments and Suggestions for Authors

The article discussess a very interesting topic. The experimental plan is well organized. After a little review, the work deverses to be published. The manuscript has been corrected in terms of English writing, results, figures, discussion, and conclusion. 

Line 85: For bacterial count, the MRS........What does bacterial count mean?indicate for which microrganisms MRS medium was used.  Did the incubation take place in anaerobiosis? If not why? Text was corrected in the manuscript to answer all questions: “For bacteria growth, the MRS (Man Rogosa Sharpe) medium supplemented with 100mg/L of cycloheximide was used. The plates were incubated in aerobiosis and anaerobiosis at 28 ºC. These two processes were performed for better growth of bacterial biodiversity (lactic/acetic acid bacteria). For yeast growth, the Sabouraud Agar medium supplemented with 50mg/L of chloramphenicol was used. The plates were incubated at 28 ºC. Serial dilutions of the samples of 10-3, 10-4, and 10-5 were performed for analyses, with subsequent surface plating according to APHA [18].”

Line 87: Explain the sampling method. As described may be difficult for readers to understand and difficult to replicate. The text was corrected in the manuscript: "Five different substrates were prepared for the fermentation process in triplicate. Each Erlenmeyer was added with 350 mL deionized water and 35 g of each type of sugar (refined, coconut, demerara, brown, cane molasses). Subsequently, 35 g of water kefir grains (10%) were added to the sugary substrate. The fermentation process was carried out for a period of 48 h at a temperature of 27 ºC. Samples (10 mL) were collected at times 0, 24 and 48 h of the fermentation process for physicochemical and microbiological analyses."

Line 122: Where is table 20? This has been corrected in the manuscript.

Line 150: The figure 1 shows the cell concentration of lactic acid bacteria and yeast....I reccomend editing the caption. Figure caption corrected. The Figure has also been modified and corrected.

Line 156: count above 7.00... add CFU/ml This has been corrected in the manuscript.

Line 213: Figure 2a error bars are missing in the graph. Error bars/standard deviation have been added to all Figures.

Submission Date

19 January 2023

Date of this review

29 Jan 2023 18:59:53

Reviewer 3 Report

-       The identification of lactic and acetic acids is ambiguous.

-       Add "Kefir" to the list of keywords.

-       Line 109: http://www.ncbi.nlm.nih.gov/BLAST [19]., Please add access dates for this links in the main text.

-       Line 122: “Table 20” change to “Table 1”

-       Line 184: “ (Table 1)” change to “(Table 2)”

Author Response

Comments and Suggestions for Authors

-       The identification of lactic and acetic acids is ambiguous. Lactic acid and acetic acid were separately identified and quantified by Chromatography - HPLC. This is shown in the new Figure 3 and discussed in the manuscript text.

-       Add "Kefir" to the list of keywords. Added the word "water kefir" to Keywords.

-       Line 109: http://www.ncbi.nlm.nih.gov/BLAST [19]., Please add access dates for this links in the main text. "Access date: 12/10/2022" has been added in the text and Table 2.

-       Line 122: “Table 20” change to “Table 1” This has been corrected in the manuscript.

 -       Line 184: “ (Table 1)” change to “(Table 2)” This has been corrected in the manuscript.

Submission Date

19 January 2023

Date of this review

03 Feb 2023 14:34:13

Round 2

Reviewer 1 Report

The large differences in initial pH made it not possible to interpret the results: was the effect due to large variations or different sugar sources (containing different nutrients)? The initial pH must be the same.

Author Response

Dear Reviewer,

We appreciate your corrections. Please see the attachment a response to your corrections.

Best Regards
